# Opinion: Understanding the impacts of agriculture and food systems on atmospheric chemistry is instrumental to achieving multiple Sustainable Development Goals

Amos P. K. Tai[1,2], Lina Luo[1], and Biao Luo[1]

[1]Earth and Environmental Sciences Programme, Faculty of Science, The Chinese University of Hong Kong, Hong Kong, China
[2]State Key Laboratory of Agrobiotechnology and Institute of Environment, Energy and Sustainability, The Chinese University of Hong Kong, Hong Kong, China

*Correspondence to*: Amos P. K. Tai (amostai@cuhk.edu.hk)

**Abstract.** Agriculture and food systems play important roles shaping atmospheric chemistry and air quality, most dominantly via the release of reactive nitrogen (Nr) compounds, but also via agricultural burning, energy use, and cropland and pastureland expansion. In this opinion, we first succinctly review our current understanding of agricultural and food-system emissions of Nr and other atmospherically relevant compounds, their fates and impacts on air quality, human health and terrestrial ecosystems, and how such emissions can be potentially mitigated through better cropland management, livestock management and whole food system transformation. With that, we highlight important knowledge gaps that warrant more extensive research, and argue that we scientists need to provide a more detailed, process-based understanding of the impacts of agriculture and food systems on atmospheric chemistry, including both chemical composition and processes, especially as the importance of emissions from other fossil fuel-intensive sectors is fading in the face of regulatory measures worldwide. Such knowledge is necessary to guide food system transformation in technologically feasible, economically viable, socially inclusive, and environmentally responsible manners, and essential to help society achieve multiple Sustainable Development Goals (SDGs), especially to ensure food security for the people, protect human and ecosystem health, improve farmers' livelihood, and ultimately help communities achieve socioeconomic and environmental sustainability.

## 1 Introduction

In and after the 2023 United Nations Climate Change Conference in Dubai, United Arab Emirates (UAE), commonly known as "COP28", more than 150 nations have signed the "UAE Declaration on Sustainable Agriculture, Resilient Food Systems, and Climate Action", emphasizing the desperate need to integrate agriculture and food systems into their climate action to reach the climate goals set forth in the Paris Agreement. For the first time, agriculture has come under the spotlight of international climate negotiation, showcasing the important roles food systems play in shaping climate via contributing to a third of global anthropogenic greenhouse gas (GHG) emissions (Crippa et al., 2021). Such momentum gathered is arguably also a promising development for air quality managers and policy makers worldwide, because agriculture and food systems

are major sources of various short-lived chemical species that shape chemical composition and processes in the atmosphere, which in turn contribute to air pollution.

"The food we eat, the air we breathe", the title of a recent review article (Balasubramanian et al., 2021), highlights succinctly the deep interconnection between these two things everyone needs for survival but often thinks too little about. We all need a minimum amount of nutrition from food to survive, and often a lot more for a thriving, productive and quality life. Due to population growth, rising incomes and shifting dietary habits across the world, the global food demand has increased roughly threefold from 1960 to 2010, and is projected to rise further by 40–50% by year 2050 depending on the scenario (Fao, 2018). Despite substantial gains in agricultural production to meet the rising demand due to the advancement of "Green Revolution" technologies, intensified agricultural inputs as well as cropland and pastureland expansion, undernourishment remains prevalent with a global rate of 11% in 2012; in low- and middle-income countries, the undernourishment rate can be as high as 20% in Sub-Saharan Africa and 16% in South Asia in 2012 (Fao, 2018). Even though the global food systems can indeed produce enough food for everybody, persistent poverty, inequality, uneven distribution, conflicts and socio-political instability cause people in many parts of the world to still go hungry on a daily basis. The challenge to satisfy the continuously rising food demand is further aggravated by environmental problems such as climate change and air pollution, which can severely threaten crop production and food security worldwide (Tai and Martin, 2017; Tai et al., 2014). Therefore, 193 Member States of the United Nations (UN) came together in 2015 to endorse SDG2 "Zero Hunger" as one of the 17 Sustainable Development Goals (SDGs) for the 2030 Agenda for Sustainable Development, aiming primarily to end poverty, hunger and malnutrition by year 2030, and to make the food systems more sustainable and resilient to climate change. The "UAE Declaration" mentioned above reinforced the importance of these food-centered goals for global sustainable development.

The tremendous gains in agricultural production in the past half-century have also posed severe threats to the environment, including the air we breathe. In addition to contributing to more than 30% of global GHG emissions, agricultural expansion and intensification have been a major driver of deforestation, land and water degradation, and biodiversity loss (Foley et al., 2011). The global food systems, including all the stages of pre-production, production, post-production, consumption and waste management, are estimated to account for 58% of global anthropogenic emissions of primary fine particulate matter (PM$_{2.5}$, i.e., particulate matter with a diameter of 2.5 μm or smaller), 72% of ammonia (NH$_3$), 13% of nitrogen oxides (NO$_x$ = NO + NO$_2$), 9% of sulfur dioxide (SO$_2$), and 19% of non-methane volatile organic compounds (VOCs) (Balasubramanian et al., 2021). Such emissions are estimated to be responsible for 22% of global mortality arising from poor air quality and 1.4% of global crop production losses in year 2018 (Crippa et al., 2022b). Moreover, reactive nitrogen (Nr) compounds of agricultural origins including NH$_3$, NO$_x$, nitrous acid (HONO) and their reaction products, can readily be deposited back onto the land surface and waterbodies, causing various effects on terrestrial and aquatic ecosystems, including more serious nutrient leaching, soil acidification (Guo et al., 2010; Lu et al., 2011), and eutrophication (Deng et al., 2023; Deng et al., 2024a; Jickells et al., 2017; Liu et al., 2023a). They may also enhance plant growth and soil carbon storage especially where nitrogen is a limiting nutrient (Thomas et al., 2010; Zhao et al., 2017b; Liu et al., 2021b), but such enhancements generally favor the more competitive plant species and may ultimately reduce species diversity of plant

communities (Bobbink et al., 2010). All these findings highlight the importance of agriculture and food systems in shaping atmospheric composition and chemical processes, as well as air pollution and the associated public health and ecosystem impacts.

The nitrogen load released by anthropogenic activities has also exceeded the so-called planetary boundary, meaning that human disturbances of the nitrogen cycle are destabilizing natural ecosystems to a possibly irreversible extent (Richardson et al., 2023). In a recent Nature Portfolio journals' collection on "Air Pollution and Global Solutions", out of 34 featured articles, only five directly address food-system emissions or food security issues, and all of them emphasize substantial knowledge gaps in understanding agricultural and food-system impacts on the atmospheric environment. Mitigating agricultural emissions will be even more important in the future as global air quality control efforts targeting mostly sources from the energy and transportation sectors have already substantially reduced $NO_x$ and $SO_2$ emissions in many parts of the world. But how can we do that without compromising the needs of people to be food-secured and nourished? How can we achieve these multiple goals under the concurrent threat of climate change, which can both impair crop production and elevate agricultural emissions? Here we argue that, to protect people and ecosystems from the harmful effects of air pollution worldwide but especially in developing regions, society needs to lay larger emphasis than now on reforming the food systems and mitigate their emissions of various pollutants, while ensuring food security for the people and livelihood of the farmers. To support that, scientists need to provide a more solid understanding of how different parts and stages of the food systems emit different compounds, how these compounds are transported, transformed and deposited back onto the surface, how all these processes are sensitive to climate change, and how the food systems can be modified in technologically feasible, economically viable and socially equitable manners to abate emissions.

## 2 How agriculture and food systems shape atmospheric chemistry and air pollution

Agriculture and food systems profoundly impact the atmosphere, most dominantly through the substantial emissions of reactive nitrogen (Nr) compounds from cropland and livestock systems, but also through other atmospheric pollutants such as primary particulate matter (PM), carbon monoxide (CO), methane ($CH_4$), $SO_2$, and VOCs via agricultural burning, energy use of the whole food systems, and deforestation to clear lands for agriculture. Among these compounds, $NH_3$, $NO_x$, and HONO are inherently chemically active and play significant roles in atmospheric processes, leading to the formation of air pollutants such as $PM_{2.5}$ and tropospheric ozone ($O_3$), which subsequently harm human health. Previous studies have enhanced our understanding of the mechanisms and driving factors behind agricultural emissions, allowing for improved evaluation of their impacts on air quality, human health and ecosystems (Butterbach-Bahl et al., 2013; Crippa et al., 2022a; Gu et al., 2023; Pilegaard, 2013). However, substantial uncertainties remain in these studies. Below is not meant to be a comprehensive review but is intended to highlight the key understanding, as well as the lack thereof, of the effects of agriculture and food systems on atmospheric composition and chemical processes. Figure 1 summarizes the important stages and impacts of agriculture and food systems via shaping atmospheric chemistry.

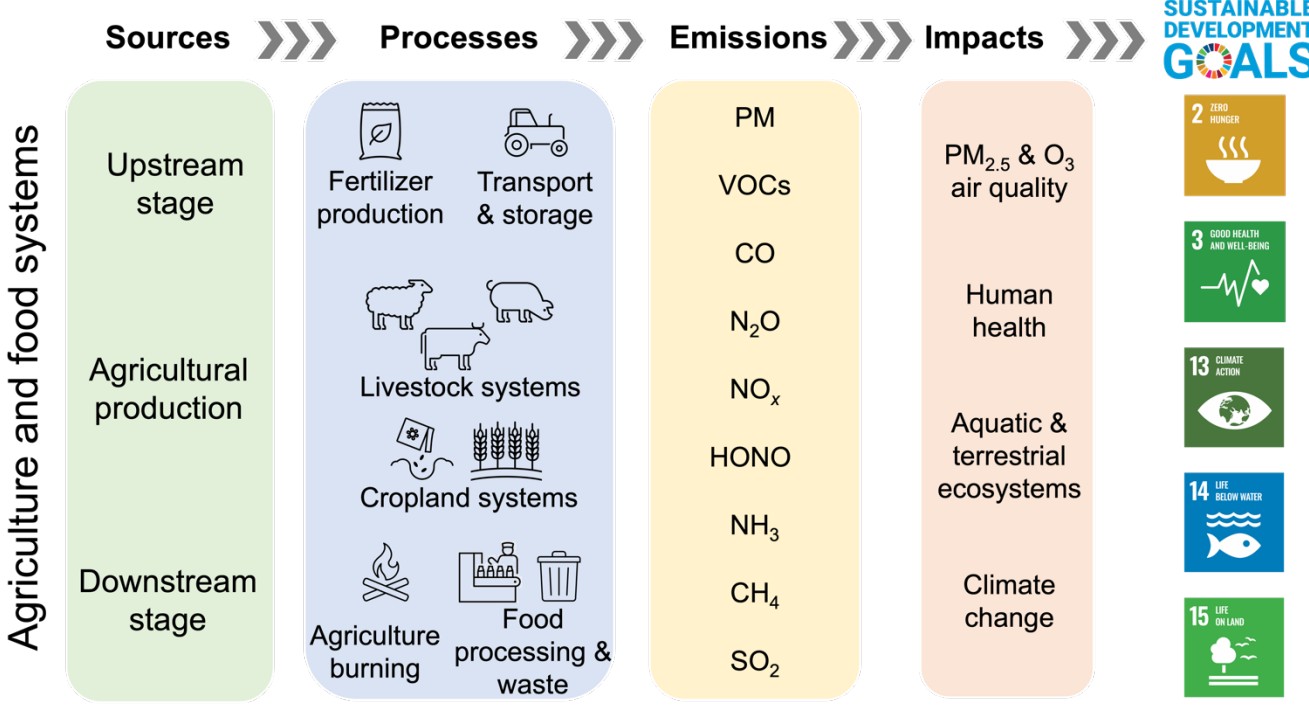

Figure 1. Effects of agriculture and food systems on atmospheric chemistry and downstream impacts on human and ecosystem health, with direct linkages to various Sustainable Development Goals.

## 2.1 Emissions of reactive nitrogen

### 2.1.1 Sources, processes, and characteristics

Nitrogen is predominantly found in its inert form, dinitrogen ($N_2$), in nature (Galloway et al., 2013). Only a small fraction of nitrogen is reactive as Nr and readily available to organisms. The advent of the Haber-Bosch process has revolutionized the way humans utilize nitrogen, allowing for the conversion of $N_2$ into $NH_3$ for fertilizer and other uses. Since then, the use of nitrogen-based fertilizers in agriculture has substantially increased, rising from 11.4 Tg N in 1961 to 108 Tg N in 2021 (Fao, 2024). This intensive and excessive use of fertilizers often surpasses crop nutrient demands, whereby only about half of the applied nitrogen is harvested in crops (Zhang et al., 2015). In livestock systems, nitrogen use efficiency (NUE) is even lower, with only 10% of the nitrogen in feed being converted to livestock products (Uwizeye et al., 2020). Consequently, in both cropland and livestock systems, a significant portion of the added nitrogen is lost to the environment after undergoing various biogeochemical processes primarily mediated by microbes, leading to the emissions of many Nr compounds. Globally, around 60% of agricultural $NH_3$ emissions are from livestock production, with the rest from cropland systems (Yang et al., 2023a).

Specifically, NH$_3$ is released through multi-stage volatilization processes. In cropland systems, NH$_3$ production follows the deprotonation of the ammonium component in fertilizers, involving urea hydrolysis, NH$_3$-ammonium equilibrium, and NH$_3$ exchange between soil and air, with higher volatilization in regions with high temperatures and alkaline soils (Freney et al., 1981). In livestock systems, NH$_3$ is primarily released during housing, storage, and spreading of manure (Webb et al., 2005). Other Nr gases such as NO$_x$ and HONO, along with the potent greenhouse gas nitrous oxide (N$_2$O), are predominantly produced from agricultural soils through microbially mediated nitrification (i.e., ammonium being oxidized to nitrate under aerobic conditions) and denitrification (i.e., nitrate being transformed to dinitrogen gas under anaerobic conditions) processes into the atmosphere (Butterbach-Bahl et al., 2013; Pilegaard, 2013). When considering the entire food systems beyond agricultural production, Nr emissions can be even higher. Food-system energy use, encompassing activities such as fertilizer production, transportation, and processing, along with land use change driven by agricultural expansion, also contributes to substantial NH$_3$ and NO$_x$ emissions (Balasubramanian et al., 2021).

Agricultural and food-system Nr emissions exhibit high spatiotemporal variations, responding nonlinearly to meteorological conditions, soil properties, and farming practices, influenced by microbial activities. Typically, regions with intensive fertilizer use and low nitrogen use efficiency (NUE, i.e., the ratio of nitrogen removed with the harvest to nitrogen input) tend to have the highest emission levels. High temperature and precipitation also contribute to increased emissions and modulate their interannual variability (Griffis et al., 2017; Shen et al., 2020). NUE and Nr emission changes can be further driven by socioeconomic factors, with divergent patterns in different countries depending on the population level, economic growth, farm size, urbanization level, international trade, and their interactions. A series of global-scale, long-term analyses have suggested that developed regions with well-managed urban-rural development tend to have lower agricultural Nr emissions as their large-scale farming along with advanced agricultural technology and coupled cropland-livestock systems can enhance NUE and maintain agricultural productivity to support international trade (Deng et al., 2024b; Gu et al., 2020; Ren et al., 2022; Liu, 2023). In the future, fertilizer input is expected to further increase to feed the growing global population, potentially further elevating Nr emissions if not efficiently managed. Meanwhile, climate change has been estimated to increase Nr emissions by ~80% between 2011 and 2100, and the resulting more frequent extreme weather events may induce extensive dry-wet and freeze-thaw cycles that can further exacerbate such increases (Griffis et al., 2017; Shen et al., 2020; Wagner-Riddle et al., 2017).

### 2.1.2 Emission estimates and associated uncertainties

Emission estimation plays a crucial role in investigating the impact of agriculture and food systems on atmospheric chemistry. Agricultural Nr emissions are typically estimated using two primary approaches: bottom-up and top-down methods. The bottom-up approach can be further categorized into multiplicative schemes based on emission factors (EFs) and mechanistic process-based models. The EF approach estimates agricultural emissions as multiplicative functions of agricultural activities (e.g., fertilizer use, livestock population) and their corresponding EFs under "standard" conditions (Misselbrook et al., 2000; Bouwman et al., 1997). Since agricultural emissions are influenced by multiple factors, including meteorological

conditions, soil properties, and farming practices, the most advanced EF methods refine their EFs by localizing these factors as much as possible. For livestock systems, refined EFs can be developed for each stage along the manure management chain (e.g., housing, storage, spreading) to achieve more accurate estimation (Huang et al., 2012). Process-based methods that rely on agroecosystem models is the most advanced bottom-up approach to estimate emissions from croplands. Agroecosystem models, such as the DayCent and Denitrification-Decomposition (DNDC) models, explicitly track the transport, biogeochemistry and fates of Nr in the soil in a mechanistic manner (Vira et al., 2020; Li et al., 2000; Del Grosso et al., 2009), and can reflect the nonlinear responses of emissions to their major drivers.

Top-down inversion methods have also been developed to refine emission estimates. This approach uses *a priori* bottom-up estimates (e.g., from EFs) and then assimilates observational data via air quality modeling to create *a posteriori* estimates, aiming to minimize discrepancies between observations and estimates. Satellite-derived observations of atmospheric $NH_3$ and $NO_x$ column concentrations have widely been used to improve agricultural emission estimates, especially in regions lacking field measurements. More recently, the launch of geostationary satellites with high spatiotemporal resolutions (e.g., TEMPO, GEMS) is expected to further enhance the accuracy of agricultural emission estimates over North America and Asia (Zoogman et al., 2017; Kim et al., 2020).

Within the air quality research community, the bottom-up approach is the most commonly used for estimating agricultural emissions. However, the derived emission estimates often exhibit high uncertainties, with variations of up to a factor of two to three, and, in some cases, even an order of magnitude different from observations (Table 1). The refinement and localization of EFs relies on extensive field measurements, which renders EF estimates relatively more accurate in heavily researched regions such as the US, Europe, and China (Ma et al., 2021; Vigan et al., 2019). However, for developing regions such as Latin America, Africa, and South Asia, sporadic field measurements are not sufficient for EF refinement and localization, where emission estimates often rely on general EFs obtained from more developed regions, lack accurate activity data (e.g., fertilizer input, manure use), and thus suffer significant uncertainties. This also contributes to the substantial differences between different global inventories. Another limitation is the relatively coarse temporal resolution (often on a monthly scale), which makes it difficult to capture the influence of abrupt increases in agricultural $NH_3$ emissions on atmospheric composition, as $NH_3$ typically peaks within several days after fertilizer application (Nelson et al., 2019).

Process-based models, often favored by the agroecosystem research community for analysis on field scales (~100 ha) and daily timescales, face challenges in regional and larger-scale applications due to their high demand for input data. Nevertheless, recent studies have successfully integrated process-based models with air quality models to estimate Nr emissions and their impacts on the atmosphere (Luo et al., 2022a; Balasubramanian et al., 2020), leading to enhanced spatiotemporal accuracy. Despite these improvements, most agroecosystem models are parameter-intensive, requiring field measurements to constrain the default parameter values. It remains questionable whether agroecosystem models can be effectively applied on larger, pan-regional scales, as even in the US and China field measurements are insufficient to cover all types of cropping systems. Furthermore, some recent studies have indicated that agricultural emissions may be substantial during non-growing seasons (Yang et al., 2022), may stem from some neglected nitrogen-cycle processes (Wrage-Mönnig et

al., 2018), and may be stimulated by dry-wetting and freeze-thawing events (Del Grosso et al., 2022). A further refinement in nitrogen-cycle representation within agroecosystem models is much warranted.

Table 1. Global estimates of $NH_3$ and $NO_x$ emissions (Tg N yr$^{-1}$).

| Sources | Method | Base year | Agricultural $NH_3$ | Total $NH_3$ | Agricultural $NO_x$ | Total $NO_x$ |
|---|---|---|---|---|---|---|
| EDGAR (Crippa et al., 2018) | Bottom-up | 2018 | 38.2 | 43.7 | 1.9 | 36.5 |
| CEDS (Mcduffie et al., 2020) | Bottom-up | 2017 | 39.2 | 51.6 | 2.3 | 37.7 |
| HTAP (Crippa et al., 2023) | Bottom-up | 2018 | 42.5 | 48.5 | 1.7 | 35.6 |
| Fowler et al. (2013) | Bottom-up | 2010 | 59.9 | 69 | | |
| Yang et al. (2023b) | Bottom-up | 2018 | 60 | | | |
| Huang et al. (2017) | Bottom-up | 2014 | | | | 39.2 |
| Luo et al. (2022b) (EDGAR as prior) | Top-down | 2018 | | 71.9 | | |
| Miyazaki et al. (2017) (EDGAR as prior) | Top-down | 2014 | | | | 47.5 |

Along the entire food supply chain, emission estimation beyond the on-farm stage generally employs a similar EF method. Uncertainties associated with these estimates primarily stem from the activities themselves, as well as from the corresponding EFs, due to the paucity of activity data. This issue is particularly profound in emissions originating from food transportation, which involves aspects such as transportation distances, means (e.g., road, rail, or ship), and refrigeration technology. International trade further complicates such estimation. Additional uncertainties arise from how the boundaries of

the food systems are defined; e.g., some studies considered the transportation of fertilizers, machinery and pesticides, while others did not (Li et al., 2022). A series of comprehensive assessment and life-cycle frameworks have been proposed recently to estimate global emissions from the entire agriculture and food systems (Crippa et al., 2022a; Li et al., 2022; Li et al., 2023). However, these frameworks still suffer uncertainties in collecting activity data and assuming different food trade policies, underscoring the need for further refinement in their methodology for emission estimation.

**2.2 Emissions of other air pollutants: Characteristics and uncertainties**

      In addition to Nr, agriculture and food systems are also major sources of a range of atmospherically relevant compounds, including PM, CO, $CH_4$, $SO_2$, and VOCs (Crippa et al., 2022a), much of which can be closely linked to agricultural

burning practices (e.g., for managing crop residues, land clearance). About 11% ($83 \pm 14$ Mha yr$^{-1}$) of total burned area globally is attributed to crop residual management, primarily occurring in South and Southeast Asia, and Sub-Sahara Africa (Chen et al., 2023). In developed countries such as the US and European nations, agricultural burning is heavily regulated, with a focus on promoting alternative methods for managing crop residues and allowing controlled burning under specific meteorological conditions to minimize environmental impacts (Hall et al., 2021; Nematian et al., 2023). In contrast, agricultural burning remains widespread in developing regions, often due to the limited time between cropping seasons and high costs of alternative management methods (Lin and Begho, 2022). Additionally, deforestation accounts for $3.8 \pm 1.2$ Mha yr$^{-1}$ of the global burned area, which is frequently observed in South America and sub-Sahara Africa (Chen et al., 2023), and is largely driven by the expansion of pasturelands and croplands (e.g., soybean and palm tree cultivation).

Agricultural burning causes significant emissions of air pollutants, e.g., representing the largest source of primary PM from agriculture and food systems (Balasubramanian et al., 2021). The estimation of agricultural fire emissions often relies on satellite-derived datasets, such as the Global Fire Emissions Database (GFED) based on the 500-m MODIS burned area product (Van Der Werf et al., 2017), which poses challenges for detecting small fires such as crop residual burning. Although the "small fire boost" method has been applied in GFED v4.1s to enhance the identification of small fires, the improvement of accuracy is limited (Zhang et al., 2018). It is thus important to devote more attention to the characterization of air pollutants from agricultural burning, not only because of their large emissions, but also because they are important considerations in formulating equitable emission reduction policy in developing regions, where the poorer agricultural populations are disproportionately affected. Finally, other practices in the food systems, including manure management and use of machinery and vehicles, contribute to the release of VOCs and $SO_2$, responsible for 16% and 12% of the total global emissions of VOCs and $SO_2$, respectively (Crippa et al., 2023). However, food-system energy use is rarely accounted for, which also limits the assessment of the impacts of the entire food systems on the atmosphere.

## 2.3 Effects on atmospheric chemistry and ecosystems

Once released into the atmosphere, agricultural and food-system emissions are actively involved in atmospheric processes and contribute to the formation of health-damaging air pollutants including $PM_{2.5}$ and $O_3$ (Fig. 1). In particular, for $PM_{2.5}$, agricultural $NH_3$ and $NO_x$ can contribute to secondary inorganic aerosols, key components of $PM_{2.5}$, which is a major health risk worldwide, responsible for millions of premature deaths annually (Lelieveld et al., 2015). As the most abundant alkaline gas in the atmosphere, $NH_3$ neutralizes sulfuric acid ($H_2SO_4$) to form ammonium sulfate and, when in excess, reacts with nitric acid ($HNO_3$) produced from the oxidation of $NO_x$ to form ammonium nitrate. Agricultural burning also contributes to $PM_{2.5}$ both as a component of primary PM and via secondary formation from emitted $SO_2$, $NO_x$ and VOCs. For $O_3$, surface $O_3$ is predominantly formed through the photochemical oxidation of CO and VOCs in the presence of $NO_x$. Agriculture influences $O_3$ formation mostly via its contribution to NO emissions. $O_3$ is either sensitive to $NO_x$ or VOCs emissions depending on whether the atmospheric chemical regime is $NO_x$-limited (i.e., low-$NO_x$ environment) or VOC-limited (i.e., high-$NO_x$ environment). Agricultural emissions ultimately influence ecosystem health as primary and secondary Nr compounds of

agricultural origins can finally be deposited back onto the surface, thus disrupting the nutrient content and cycling in the underlying ecosystems.

The significant roles that agriculture and food systems play in shaping chemical processes in the atmosphere are increasingly realized. It is estimated that they contribute to 22–53% of $PM_{2.5}$ and 5–25% of $O_3$ pollution, which are contributions comparable to those of other well-regulated sources driven by fossil fuel combustion such as the energy and transportation sectors (Crippa et al., 2022a). However, there is still a lack of thorough investigation, particularly in underdeveloped regions such as Africa and South Asia. In this section, we highlight the latest findings along with the uncertainties and limitations associated with the impacts of agriculture and food systems on the atmospheric environment.

### 2.3.1 Impacts of agricultural $NH_3$ emissions on $PM_{2.5}$ pollution and human health

Agricultural $NH_3$ emissions significantly contribute to $PM_{2.5}$ pollution. Traditional $PM_{2.5}$ control policies have targeted mainly combustion-related emissions of $SO_2$ and $NO_x$, which have already led to significant improvements in $PM_{2.5}$ air quality in regions such as the US, Europe and, more recently, China, but ongoing efforts are still essential for further air quality improvements especially in developing countries, but even in cleaner regions such as the US that still witnesses thousands of deaths every year (Thakrar et al., 2020; Tschofen et al., 2019). More importantly, $NH_3$, another important precursor of $PM_{2.5}$, has historically received much less attention, and its primary source, agriculture, has always been less regulated than other sectors. However, agriculture $NH_3$ is an increasingly important contributor to $PM_{2.5}$ globally, accounting for approximately 34% of annual $PM_{2.5}$ concentrations in Europe, 23% in the western US, 36% in the eastern US, and 31–33% in China (Bauer et al., 2016; Han et al., 2020; Pozzer et al., 2017). The dominant influence of $NH_3$ on $PM_{2.5}$ is via affecting ammonium nitrate formation, especially during winter (Han et al., 2020; Pozzer et al., 2017).

Due to the strong nonlinearity of inorganic aerosol chemistry, the sensitivity of $PM_{2.5}$ to $NH_3$ emissions varies widely across different regions, mostly depending on the regional atmospheric conditions, seasonal meteorological conditions, and the intensity of mitigation efforts (Thunis et al., 2021). The $PM_{2.5}$ burden in China shows a higher sensitivity to agricultural $NH_3$ emissions compared to combustion-related $NO_x$ emissions (Bauer et al., 2016). For the Beijing-Tianjin-Hebei region in particular, a joint control of $NH_3$, $NO_x$, and $SO_2$ is essential, especially as $NO_x$ and $SO_2$ levels remain high (Fu et al., 2017; Liu et al., 2021c). In the western US, $PM_{2.5}$ sensitivity to $NH_3$ reductions is pronounced with reduction intensities of 40% to 60% (Bauer et al., 2016). In the eastern US and India, $PM_{2.5}$ shows similar sensitivities to both combustion $NO_x$ and agricultural $NH_3$, while Europe demonstrates greater sensitivity to $NO_x$ than $NH_3$ emissions, particularly in western Europe, but a joint control strategy is preferred in eastern Europe (Bauer et al., 2016; Liu et al., 2023b).

Furthermore, some studies have directly examined the health damage related to air quality associated with crop production processes, highlighting that animal-based foods contribute to higher PM pollution and subsequent health damage than plant-based foods, as livestock management results in greater $NH_3$ emissions compared to fertilizer applications on croplands (Domingo et al., 2021). Health effects induced by fertilizer use are more significant in densely populated regions close to the farms (Hill et al., 2019).

In general, despite lower $NH_3$ emissions at lower temperatures, the effects of mitigating agricultural $NH_3$ are stronger in winter, when lower temperatures favor the formation of ammonium nitrate (Pozzer et al., 2017). $PM_{2.5}$ formation is sensitive to reductions in $NO_x$ emissions in $NH_3$-rich environments and becomes more sensitive to $NH_3$ in environments with lower $NH_3$ levels (Liu et al., 2023b; Holt et al., 2015; Ansari and Pandis, 1998). The relative effectiveness of controlling agricultural $NH_3$ emissions may diminish when substantial amounts of $NO_x$ and $SO_2$ are under control, as $NH_3$ is more likely to remain in the gas phase rather than contributing to $PM_{2.5}$ formation (Fu et al., 2017). Controlling agricultural emissions benefits not only rural areas but also downwind urban regions especially for poorer populations near the farms (Hill et al., 2019). From a policy-making perspective, $NH_3$ abatement may be even more cost-effective than $NO_x$ for controlling $PM_{2.5}$ pollution (Gu et al., 2021; Pinder et al., 2007).

Uncertainties and limitations still abound in our understanding of the impacts of agricultural $NH_3$ on $PM_{2.5}$ formation. A major source of uncertainty stems from the nonlinear and high sensitivity of $PM_{2.5}$ to the nitrate-ammonium ratio, which may be prone to large errors due to uncertainties in both $NO_x$ and $NH_3$ emission estimates. Consequently, even within the same region, the response of $PM_{2.5}$ to agricultural $NH_3$ emissions can vary between studies. For instance, one study suggested that reducing agricultural $NH_3$ emissions by 40% could decrease secondary inorganic PM in winter haze events by 21% (Han et al., 2020), while another found that a reduction of over 50% was needed to have similar effects in the same region (Guo et al., 2018; Song et al., 2019). Another source of uncertainty lies in the source apportionment methods used to estimate the contribution of agricultural emissions to $PM_{2.5}$. Source apportionment studies relying on air quality models use either the brute force method (BFM, also known as the zero-out method) or the tagged species-based approach. A recent study applying both methods to estimate the impact of agricultural $NH_3$ emissions found that the tagged species-based method attributes a 16% contribution to $PM_{2.5}$, whereas estimates from BFM reach up to 33% (Han et al., 2020). While BFM is effective for sensitivity analysis in examining the responses of $PM_{2.5}$ to reductions in precursor emissions, the tagged species-based method is more suitable for source contribution studies owing to the nonlinear nature of $PM_{2.5}$ to its precursors.

From the perspective of $PM_{2.5}$ pollution control, region-specific investigation for the responses of $PM_{2.5}$ to precursor reductions with higher spatial resolutions is strongly preferred to larger-scale (e.g., national) analysis, and such investigation should be updated periodically as emission inventories are revised. There is also a lack of studies with high temporal resolutions, such as weekly or daily, which is particularly important because $NH_3$ emissions typically peak about one week after fertilizer application and such temporal details may influence episodic $PM_{2.5}$ pollution but are lost if monthly emissions are used (Nelson et al., 2019). A more detailed spatiotemporal analysis shall refine our understanding of the specific locations and periods most influenced by agricultural activities, possibly enabling more effective pollution mitigation strategies.

### 2.3.2 Impacts of agricultural burning on air quality and human health

Agricultural burning significantly shapes atmospheric chemistry, particularly in South Asia and Africa, leading to the formation of harmful air pollutants including $PM_{2.5}$ and ozone ($O_3$), mostly via substantial emissions of primary PM, CO, $CH_4$, and VOCs. Once these pollutants are released into the atmosphere, they can affect not only local areas but also be transported

to downwind regions. The pollution-related health burdens from agricultural burning disproportionately affect low-income individuals in rural areas or near the burning sites (Reddington et al., 2021). $PM_{2.5}$ emissions from agricultural fires are often considered more harmful than those from other sources due to their composition and the potential for long-range transport (Lin and Begho, 2022). Specifically, in Delhi, India, agricultural burning is shown to be responsible to approximately 7% to 78% of the enhanced $PM_{2.5}$ concentrations (Cusworth et al., 2018). In Southeast Asia, agricultural and deforestation fires are estimated to account for about 40% to 70% of annual $PM_{2.5}$ concentrations in northern Thailand, Myanmar, Cambodia, and Laos, resulting in ~59,000 annual premature deaths (Reddington et al., 2021). These fires also contribute to $O_3$ pollution, accounting for 5% of the average daily maximum 8-hour $O_3$ concentration and causing ~3,800 annual premature deaths (Reddington et al., 2021). Agricultural burning is a major source of $PM_{2.5}$ pollution in South Asia, contributing to its status as one of the most polluted regions globally (Lan et al., 2022; Lin and Begho, 2022). In Africa, agricultural burning contributes to 22% of the annual average $PM_{2.5}$, leading to 106,000 premature deaths, though another study estimated a lower number of ~43,000 deaths (Gordon et al., 2023).

Agricultural burning in South Asia, Southeast Asia and Africa is challenging to detect and characterize quantitatively due to its small scales but large numbers, and estimates based on satellite observations suffer from inadequate resolutions for such detection, leading to significant uncertainties in emission estimates (Korontzi et al., 2006). In addition, the widely-use bottom-up methods for emission inventories heavily rely on crop-type specific EFs, but often use fixed factors for different crops, further increasing uncertainties (Zhang et al., 2020a). The lack of air monitoring networks in these regions further complicates the linkage between fire activities and pollution-related health damage. More field measurements to identify important emitted species and track their chemical transformation for different cropping systems or crop types, especially in developing regions, are very much warranted.

### 2.3.3 Impacts of agricultural $NO_x$ and HONO emissions on air quality

Agricultural emissions of $NO_x$ from fertilized soils, historically overlooked in $O_3$ research, are now acknowledged for their impacts on $O_3$ pollution in agriculturally intensive regions. Recent studies in rural areas with intensive agricultural activities have shown that $NO_x$ emissions from fertilized soils significantly enhance ozone formation (Romer et al., 2018). For example, in California, agricultural NO emissions account for approximately 40% of the total $NO_x$ emissions and contribute to ~23% to $O_3$ formation (Sha et al., 2021). In China, agricultural soil $NO_x$ emissions may also account for ~40% of $O_3$ nonattainment in some regions of China (Huang et al., 2023). Similarly, a US study suggested that in low $NO_x$ environments, controlling agricultural soil $NO_x$ emissions is more effective for $O_3$ reduction than the same level of control on biogenic VOCs (Geddes et al., 2022). Beyond $NO_x$-limited regions, agricultural $NO_x$ emissions are also influential in some $NO_x$-saturated or transition-regime areas where agricultural $NO_x$ emissions are on a par with combustion-related $NO_x$ emissions; there controlling agricultural $NO_x$ emissions can be more effective than other anthropogenic sources (Lu et al., 2021b).

As the mitigation of anthropogenic non-agricultural $NO_x$ emissions become more successful, many regions may eventually transition to being $NO_x$-limited, suggesting that the importance of agricultural $NO_x$ to $O_3$ control is expected to rise.

A better understanding of the impacts of agricultural NO$_x$ on O$_3$ chemistry requires more accurate emission estimation and more precise source apportionment analyses. Previous studies using air quality models have either neglected soil NO$_x$ emissions or relied on simplified EF-based methods that fail to capture spatiotemporal variability of emissions. Recent advancements in mechanistic parameterization schemes, such as the Berkeley-Dalhousie Soil NO$_x$ Parameterization (BDSNP) scheme (Hudman et al., 2012), have improved our understanding of soil NO$_x$ and O$_3$ chemistry, but more field measurements from poorly researched regions are much needed to enhance regionalized applicability.

Finally, beyond NH$_3$ and NO$_x$, agricultural emissions of HONO are also important for atmospheric chemistry by affecting chemical processes in the atmosphere, mostly because of its photolysis product, hydroxyl radical (OH), the primary oxidant in the troposphere, which is heavily involved in PM$_{2.5}$ and O$_3$ chemistry (Oswald et al., 2013). A recent modeling study revealed that HONO emissions from fertilized agricultural soils could increase average daytime O$_3$ and daily particulate nitrate concentrations across the North China Plain by 8% and 47%, respectively (Wang et al., 2021), and by 4.6% and 14%, respectively, even in non-growing seasons (Wang et al., 2023). However, more accurate parameterization for HONO emissions is needed to improve the estimates.

### 2.3.4 Impacts of nitrogen deposition on terrestrial and aquatic ecosystems

The Nr compounds of agricultural origins often undergo transport and chemical transformation, and are eventually deposited back onto the surface of terrestrial and aquatic ecosystems, resulting in increased nitrification, nutrient leaching, soil acidification (Guo et al., 2010), eutrophication (Liu et al., 2023a), and biodiversity loss (Simkin et al., 2016), while also possibly enhancing forest growth and carbon storage (Liu et al., 2022; Lu et al., 2021a; Quinn Thomas et al., 2010) as well as marine productivity (Jickells and Moore, 2015). Enhanced Nr deposition to the open ocean has been known to generate high-productivity, low-oxygen zones with disrupted ecosystem functions (Doney, 2010). Due to historically more stringent emission controls on combustion NO$_x$ than agricultural NH$_3$ emissions, Nr deposition patterns are shifting from being nitrate-dominated to ammonium-dominated, a trend observed in the US and China, and expected in Europe (Chen et al., 2020; Li et al., 2016; Liu et al., 2020; Tan et al., 2020), not only over inland but also in coastal areas (Liu et al., 2023a). Although the deposition of oxidized Nr compounds has decreased, increased deposition of reduced Nr compounds from agricultural NH$_3$ emissions, particularly in regions with intensive fertilizer use or near animal feeding operations, may offset such reduction (Chen et al., 2020; Tan et al., 2020). Control measures for Nr deposition show varied effectiveness between oxidized and reduced forms. Each unit of NO$_x$ control can achieve 80–120% reductions in oxidized deposition, whereas each unit of NH$_3$ control can only achieve 60–80% reductions in reduced-form Nr deposition (Tan et al., 2020).

A recent paper has systematically reviewed the quantification methods for nitrogen deposition and summarized the major uncertainties (Zhang et al., 2021a). Global monitoring networks for nitrogen deposition have been established, especially in the US, Europe, and East Asia, offering relatively accurate data for wet deposition. However, significant challenges remain in measuring dry deposition due to the need for highly advanced instruments and analysis methods. Further technological innovations in measurements are warranted. The spatial distribution of observation sites also needs to be optimized to cover

more representative locations and reduce sampling time to prevent sample losses. In addition, integration of Earth system models and satellite retrievals has enhanced our understanding of the spatial distribution and temporal variations of Nr deposition and their ecosystem effects (Liu et al., 2017; Zhao et al., 2017a). Nevertheless, model estimation of Nr deposition still has substantial limitations that arise from poor representation of the bidirectional exchange of $NH_3$, inaccurate dry deposition velocities, poor representation of organic nitrogen compounds, and uncertainties in Nr emission estimates. The utility of satellite observations is also constrained by their spatiotemporal coverage and retrieval methods. To enhance our current understanding of Nr deposition, a comprehensive framework that integrates these methods, supported by international collaboration, is strongly encouraged.

## 3 How agriculture and food systems can be transformed to mitigate emissions

### 3.1 Cropland systems

Nitrogen management in croplands is a crucial challenge of the 21$^{st}$ century, as we need to balance food production with pollution mitigation (Houlton et al., 2019; Davidson et al., 2015). To that end, NUE is a vital metric. The current global average NUE stands at ~0.4, yet we need to increase it to ~0.7 by 2050 to meet the growing global food demand while minimizing environmental degradation, in line with the UN SDGs (Alexandratos and Bruinsma, 2012; Zhang et al., 2015). NUE varies globally, with higher values in high-income countries such as the US and Canada (0.68) as well as Europe (0.52), and lower in middle-income countries such as China (0.25) and India (0.30) (Zhang et al., 2015). In low-income regions, such as Sub-Saharan Africa (0.72), NUE is initially high due to low fertilizer use but is expected to decrease as fertilizer use increases (Zhang et al., 2015).

Agricultural Nr emissions are closely tied to farming practices aimed at boosting crop productivity. The goal of nitrogen management is to match nutrient supply with crop demands effectively. Therefore, choosing appropriate farming practices, particularly adhering to the principles of "4R nutrient stewardship" (i.e., applying fertilizer with the right source, right rate, right time, and right place) (Bruulsema et al., 2009), has shown potential in mitigating Nr emissions while maintaining or even enhancing crop productivity (Gu et al., 2023). Additionally, using enhanced-efficiency fertilizers (Akiyama et al., 2009; Qiao et al., 2015) such as slow-release and controlled-release fertilizers, fertilizers containing nitrification inhibitors (NIs) and/or urease inhibitors (UIs), adopting efficient irrigation practices (Holcomb et al., 2011), and incorporating biochar amendments (Luo et al., 2023), can also help reduce Nr emissions (summarized in Table 2).

Table 2. Effects of different management strategies on agricultural nitrogen emissions, as adapted from (Gu et al., 2023).

| Strategies | | $NH_3$ | $NO_x$ |
|---|---|---|---|
| Fertilizer management | Rate | –42% | –26% |

| | | |
|---|---|---|
| Type | −66% | −37% |
| Time | +17% | −74% |
| Placement | −72% | / |
| Irrigation management | −36% | −93% |
| Biochar amendment | +38% | −19% |
| Enhanced-efficiency fertilizers | −70% | −46% |

To mitigate agricultural burning emissions. eco-friendly crop residue management options have been explored. In-situ methods such as reduced tillage hold much promise, yet they can also stimulate Nr emissions under certain conditions (Lin and Begho, 2022). Another approach involves converting crop residues into biochar or harnessing crop residues for renewable energy sources; however, these methods come with additional costs and technological requirements, making it less feasible in some developing countries (Lin and Begho, 2022). Effective crop residue management in South Asia and Africa remains a complex challenge that requires addressing various hurdles.

Cropland nitrogen management, while extensively researched, lacks a one-size-fits-all solution due to the diversity of cropping systems. The impact of various practices on Nr emissions varies significantly across regions and species. Managing Nr emissions often leads to trade-offs among different Nr species from fertilized soils (Gu et al., 2023; Pan et al., 2022; Qiao et al., 2015). Additionally, management strategies should account for other Nr losses, such as surface runoff, leaching, and potential changes in crop yield. Customized, region-specific, and even farm-specific evaluation is essential for harmonizing agricultural and environmental goals. Additionally, future climate change is likely to increase the occurrence of extreme weather events, imposing additional demands on cropland systems for resilience. This will further complicate nitrogen management, necessitating adaptive strategies to maintain agricultural productivity while managing Nr emissions effectively in the face of these evolving environmental challenges.

## 3.2 Livestock systems

Sustainable livestock management serves as another crucial pillar in achieving low-emission agriculture and food systems, with the pathway to this goal fundamentally rooted in the optimization of resource use efficiency. Guiding by this principle, a series of measures about livestock management (e.g., sustainable intensification, animal health, and recoupling between cropping and livestock systems) and manure management (technological options at feeding, housing, storage stages) have been taken. Current estimates of emission reductions from these measures are limited, leaving great uncertainties in the outcomes of currently reported mitigation measures such as those listed in Table 3. Herd size can also help improve resource use efficiency (Fao, 2023). Industrial and intensive livestock farms can produce animal products more efficiently and have lower emissions compared to small farms (Herrero et al., 2013), but focusing solely on improving resource use efficiency may compromise other aspects, such as causing local nutrient overload (Bai et al., 2022) and harming animal welfare (Fao, 2023). Furthermore, increasing industrial livestock farms disrupts nutrient recycling between livestock and croplands, inducing

nutrient imbalances (Jin et al., 2020). It is important to note that a single measure may effectively control certain air pollutants while potentially increasing other air pollutants or greenhouse gas emissions. For example, anaerobic digestion, a biological process where bacteria degrade organic matter without oxygen, produces biogas for renewable energy, which can supply on-farm energy needs with lower emissions, but may raise the $NH_3$ emissions of the digestate (Yan et al., 2024). Overall, there are a number of abatement options, but more knowledge about their effectiveness, cost-effectiveness performance, and trade-offs is required to underpin the development of abatement measures or design of sustainable livestock systems.

Table 3. $NH_3$ emission abatement efficiency for different manure management options, as adapted from Hou et al. (2015) and Zhang et al. (2020b).

| Stage | Measure | Reduction in $NH_3$ emissions |
|---|---|---|
| Feeding | Low-crude protein feeding | 24–65% |
| | Dietary additives | 33–45% |
| Housing | Floor adaption | 10–50% |
| | Frequent manure removal | 25–30% |
| | Rapid manure drying | 70–90% |
| Storage | Solid-liquid separation | 20–30% |
| | Manure surface covers | 50–88% |
| | Acidification by additives | 18–70% |
| | Composting (aeration, turning, compaction) | 55–97% |

## 3.3 Whole food systems

The entire food systems include not only on-farm production but also upstream and downstream stages such as agricultural input (e.g., fertilizer, pesticide) production, food processing, distribution, storage, retail and consumption. Emission estimation and mitigation strategies for these off-farm stages as well as along the whole food chain are further complicated by dietary changes and food loss and waste, which can affect emissions at any stage along the whole food chain. The widespread dietary shifts from plant-based to meat-intensive diets are the key driver for the globally increasing food demand, and meat-intensive diets are not only linked to increased risks of cardiovascular diseases, cancers, and type-2 diabetes, but also pose severe environmental threats (Gbd, 2019; Liu et al., 2021a). For instance, during 1980–2010 in China, dietary change alone could raise $NH_3$ emissions by 63% and annual mean $PM_{2.5}$ by up to ~10 $\mu g\,m^{-3}$ (Liu et al., 2021a). The study further suggested that adopting more sustainable, healthier, less meat-intensive diets could decrease annual mean $PM_{2.5}$ by 2–6 $\mu g\,m^{-3}$ in China. Likewise, a worldwide shift to plant-based diets could cut agricultural emissions significantly, by 44–86%, especially in regions with extensive livestock production (Springmann et al., 2023). Such dietary changes are expected to lower

PM$_{2.5}$ and O$_3$ pollution by 3–7% and 2–4%, respectively, reduce premature mortality by 3–6%, and enhance economic output by 0.5–1.1%. However, a recent detailed study on alternative dietary shifts argued that specific changes should be made cautiously, as some types of shifts aimed to improve health and nutrition may increase emissions (Guo et al., 2022). Dietary shifts toward a more plant-based diet, which encourage more intake of fruits, vegetables, and dairy products, can sometimes increase Nr emissions if such shifts require higher fertilizer inputs in low-NUE croplands.

In addition, food loss typically occurs in the pre-production and production stages due to inadequate management and technology, whereas food waste happens during retail and consumption. About one third of the total food production (~1.3 billion tonnes) is discarded as food loss and waste (FLW) (Shafiee-Jood and Cai, 2016). Efforts to reduce FLW have shown promising results in mitigating NH$_3$ emissions and PM$_{2.5}$ pollution, with estimates suggesting a potential reduction of up to 11.5 Tg in NH$_3$ emissions and a decrease of about 5 μg m$^{-3}$ in PM$_{2.5}$ levels worldwide (Guo et al., 2023). In relation to nutrition demand, populations with excessive calorie intake are recommended to shift their diets toward healthier nutritional patterns, which can also reduce FLW and emissions (Lopez Barrera and Hertel, 2023).

Overall, reducing FLW and dietary changes can have multiple benefits for people, prosperity and the planet; specifically concerning atmospheric chemistry, they help mitigate pollutant emissions throughout the whole food supply chain by reducing the overall food demand. Currently, agricultural emission abatement is usually more focused on on-farm production and the food supply. Stronger emphasis on whole food system transformation and formulating integrated policies that target both the demand and supply sides of food and agriculture is much warranted.

## 4 How the science of agriculture-environment interactions contributes to sustainable development

Sustainable development is development that aims to meet the needs of the present generation without compromising the ability of future generations to meet their own needs (Brundtland, 1987). It is a holistic approach that emphasizes that the "needs" of every one but especially the poor and disadvantaged should be prioritized, and that there are "limitations" to the environment's ability to meet such needs. The goals of sustainable development are thus to seek economic prosperity for the people in a way that is socially inclusive and environmentally responsible; that is, economy, society and environment are equally important considerations when pursuing long-term human development. The 17 UN SDGs adopted in 2015 provide a framework for governments, businesses and civil society to work toward sustainable development across all sectors, of which agriculture and food systems are among the most important, as most obviously indicated by SDG 2 "Zero Hunger", which aims to end hunger, achieve food security, enhance nutrition, and promote sustainable and climate-resilient food systems. However, the SDGs are not meant to be standalone objectives, but are interconnected and need to be considered holistically to achieve various objectives, and here we argue that a better understanding of agricultural and food-system contributions to atmospheric chemistry, including both the composition and chemical processes of the atmosphere, is indeed crucial to help stakeholders achieve SDG 2 in synchrony with other SDGs, especially SDG 3 "Good Healthy and Well-being", SDG 13 "Climate Action", SDG 14 "Life Below Water", and SDG 15 "Life on Land", but also various others more indirectly.

The previous sections have highlighted how agriculture and food systems are important sources of Nr and other air quality-relevant compounds, and thus contribute substantially to PM$_{2.5}$ pollution, and to O$_3$ pollution to a lesser but increasingly important extent. These pollutants are shown to cause some of the most fatal non-communicable diseases such as cardiovascular diseases, cancers and respiratory diseases, taking significant tolls on human health and well-being worldwide. Therefore, better understanding and quantification of these sources are crucial to achieving SDG 3 "Good Health and Well-being", which aims to ensure healthy lives and promote well-being for all at all ages. This is particularly important now as so many emission control efforts have already been in place for decades to reduce non-agricultural sources of air pollutants, such as combustion-derived NO$_x$ and SO$_2$ from the energy and transportation sectors, but relatively little has been done to mitigate agricultural emissions, and we foresee the increasing dominance of agricultural Nr as well as unmitigated agricultural burning in shaping future aerosol chemistry. To that end, as reviewed above (Sect. 2.1), a better understanding of the magnitudes and drivers of Nr emissions is much needed, and scientists need specifically to 1) conduct more field measurements in representative agricultural systems to better capture the responses of Nr emissions to driving factors and provide more comprehensive datasets for evaluating, calibrating, and refining emission models and estimates; 2) refine EFs within the EF approach by incorporating localized adjustments based on extensive field measurements for different crop types, cropping systems and livestock systems across diverse regions; 3) incorporate process-based agroecosystem models with enhanced representation of the nitrogen cycle to improve emission estimates on fine spatiotemporal scales and to capture the episodic and dynamics responses of Nr emissions to fertilizer applications, extreme weather events, and changes in farming practices; 4) utilize geostationary satellite observations with more sophisticated retrieval methods, especially for agricultural burning detection and short-term soil responses to fertilizer applications. These improvements would greatly help decrease uncertainties associated with Nr emission estimates and their adverse impacts on the atmospheric environment, and thus help us devise better control policies. Currently, only the European Union has established NH$_3$ emission control targets, aiming to reduce NH$_3$ emissions by 19% in 2030 compared to 2005 levels, as per the National Emission Ceiling Directive (Eu, 2016). In China, in late 2023, a decade after the initial launch of the Action Plan for Fighting for a Blue Sky, new actions were announced to focus on controlling agricultural NH$_3$ emissions (Council, 2023). These included specific targets for the Beijing-Tianjin-Hebei region, aiming for a 5% reduction by 2025 compared to 2020 levels. Further improvements are still needed in Europe, China, and also the US where no specific mitigation targets have been planned, but relatively extensive research in these regions has already informed policy approaches elsewhere. Other countries and regions are expected to follow suit, and more research for especially poorly researched, developing regions such as those in South Asia, Southeast Asia, Africa and Latin America are necessary to guide their mitigation efforts.

To mitigate agricultural and food-system emissions of Nr and other pollutants, in light of the complex region- and species-specific responses of Nr emissions across multiple stages from the whole food systems (Sect. 3), we also need to focus more research efforts on the various mitigation pathways, including: 1) identifying strategies that can effectively mitigate multiple Nr species and benefit agricultural productivity without exacerbating other Nr losses, acknowledging the trade-offs commonly observed in mitigation strategies; 2) developing customized strategies tailored to the specific conditions of each

region, farm or facility, given the variability in the effectiveness of Nr emission control strategies; 3) enhancing our understanding of the costs and outcomes of various mitigation measures, which is crucial for developing socioeconomically sound strategies with minimized additional investment, particularly for low-income regions; 4) emphasizing integrated policies that consider the entire food supply chain and food demand to maximize the socioeconomic and environmental benefits of emission reduction measures. Such efforts are recommended for both developing and developed regions.

Greater research efforts in the above can arguably help us address SDG 13 "Climate Action" as well, which aims to take urgent action to combat climate change and its impacts, as many of the short-lived Nr species share common sources with $N_2O$, the third most potent greenhouse gas. Furthermore, agriculture influences ecosystems not only via atmospheric Nr deposition but also via direct nutrient leaching and runoff to waterbodies. Nitrogen pollution can bring tremendous disruptions to terrestrial and aquatic ecosystems, often modifying both ecosystem productivity and biodiversity. Therefore, mitigating agricultural and food-system emissions also helps us strive toward SDG 14 "Life Below Water", which aims to conserve marine and coastal ecosystems, and sustainably use their resources for sustainable development, and SDG 15 "Life on Land", which aims to protect, restore, and sustainably manage terrestrial ecosystems, promote biodiversity conservation, and combat desertification and land degradation. Although the fates of various agricultural Nr compounds in terrestrial and aquatic ecosystems may be more within the realm of biogeochemistry, atmospheric scientists are necessary to better quantify the ecosystem input of Nr via atmospheric deposition (Sect. 2.3.4), especially via (Zhang et al., 2021b): 1) enhancing the Nr deposition monitoring network with a focus on technological innovations for dry deposition measurements and increased spatial resolutions by including more representative sites; 2) improving model-based analysis by better parameterizing both wet and dry deposition processes, as well as by providing more accurate Nr emission estimates to drive model simulations; 3) advancing satellite-based analysis with more refined retrieval methods; 4) developing a comprehensive framework that integrates monitoring, air quality modeling, and satellite observations.

It is essential to consider the mitigation strategies discussed above in synergy with other socioeconomic objectives. For instance, if top-down approaches are used to reform the food systems in ways that ignore the actual needs of the farmers, or even deprive the farmers of their livelihood, cultural heritage and social inclusion, such approaches do not abide to the tenets of sustainability even if they are effective in abating food-system emissions. Indigenous knowledge, cultures and traditions in the local food systems always have to be proactively considered. Often reducing food-system emissions would bring immediate health benefits to the farmers and people in agricultural regions in general due to the reduced exposure to airborne and waterborne (e.g., fertilizer, pesticide and animal waste runoffs) agricultural pollutants, which would in the long term improve their productivity and livelihood. Furthermore, by promoting sustainable agricultural practices, supporting local food production, improving distribution networks and reducing food waste, food system transformation can help both rural and urban populations gain access to safe, nutritious and affordable food, which is essential for fostering socially inclusive communities. Therefore, transforming the food systems in economically feasible, socially equitable and environmentally responsible manners, facilitated by better understanding of the science of agriculture-environment interactions behind, can also help us address SDG 1 "No Poverty", which aims to end poverty by addressing its root causes, promoting social protection

systems, and enhancing access to basic services and resources; SDG 6 "Clean Water and Sanitation", which aims to ensure universal access to clean water and sanitation, improve water quality and promote sustainable water management practices; and SDG 11 "Sustainable Cities and Communities", which aims to make cities and human settlements inclusive, safe, resilient and sustainable.

We therefore opine that, in consideration of the substantial impacts of agricultural and food-system emissions on atmospheric chemistry, air pollution and subsequently on terrestrial and aquatic ecosystems, we as a society need to take concrete actions to transform the food systems, so as to simultaneously ensure food security for the masses, lessen the human health and ecological impacts of agricultural pollutants, improve the livelihood of farmers and agricultural workers, and help cities and communities become economically, socially and environmentally sustainable. That is, in essence, to achieve multiple

SDGs. To that end, scientists play vital roles in providing the detailed process-based understanding of agricultural and food-system emissions as well as the fates and wider impacts of the emitted compounds. Above we have specifically highlighted several knowledge gaps and aspects that warrant much more research efforts, which are necessary to guide food system transformation along technologically and economically feasible as well as socially and environmentally responsible paths. This could be one of the key ways through which we scientists can fulfil not only our professional responsibility, but also our

social responsibility.

## Contributions

APKT conceived, wrote and revised the opinion paper, and diagnosed connections of atmospheric chemistry to sustainable development. LL and BL reviewed current literature and drafted most parts on emissions, chemistry and mitigation methods. LL was involved extensively in the revision of the manuscript.

## Competing interests

At least one of the (co-)authors is a member of the editorial board of Atmospheric Chemistry and Physics.

## Acknowledgments

This work was supported by the General Research Fund (project no.: 14307722) granted by the Research Grants Council (RGC) to APKT, as well as funding from the State Key Laboratory of Agrobiotechnology and Innovation and Technology

Commission (project no.: 8300031, 8300036, 8300070) granted to APKT.

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
