# Peer review of "Opinion: Understanding the impacts of agriculture and food systems on atmospheric chemistry is instrumental to achieving multiple Sustainable Development Goals"

_EGUsphere, 2024_

## Author Response (AR1)

Author Responses to Referees' Comments on **"Opinion: Understanding the impacts of agriculture and food systems on atmospheric chemistry is instrumental to achieving multiple Sustainable Development Goals"** by Amos P. K. Tai et al. (MS No.: egusphere-2024-293)

We would like to thank the reviewers for the thoughtful and insightful comments for this Opinion article. The manuscript has been revised accordingly, and our point-by-point responses are provided below. The referees' comments are *italicized*, our new/modified text is highlighted in **bold**.

**Response to Referee #1**

*The authors have put together a commanding review of the impacts of agriculture on the atmosphere. As such it will be a useful contribution to literature, but I recommend that the following comments be taken into consideration to further strengthen the work.*

> We appreciate the reviewer's positive feedback on our efforts to review the current understanding of the impact of agriculture and food systems on atmospheric chemistry. We also acknowledge reviewer's suggestions regarding the structure and content of this opinion article, especially the part about improving the readability for non-experts. We have revised the paper accordingly to address the reviewer's concerns point by point, with detailed responses provided below.

1. *1) The causal linkages from the source and activity emitting pollutants to the fundamental governing chemistry and impacts can be strongly and sequentially highlighted.*

> Thank you for this valuable suggestion on strengthen the casual linkages from sources and emissions to impacts, especially when these complex linkages may not be readily familiar by non-experts, and may potentially leading to some confusion in understanding the impacts of agriculture and food systems on atmospheric chemistry.

> To address this, we have added an introductory paragraph at the beginning of the second section. This paragraph outlines the overall picture of sources, emissions, and impacts without delving into excessive details, as the key points are thoroughly discussed in the subsequent sections. Additionally, we included a diagram to aid in visualizing these connections for a clearer understanding.

> L82 P3: "**Agriculture and food systems profoundly impact the atmosphere, most dominantly through the substantial emissions of reactive nitrogen (Nr) compounds from cropland and livestock systems, but also through other atmospheric pollutants such as primary particulate matter (PM), carbon monoxide (CO), methane ($CH_4$), $SO_2$, and VOCs via agricultural burning, energy use of the whole food systems, and deforestation to clear lands for agriculture. Among these compounds, $NH_3$, $NO_x$, and HONO are inherently chemically active and play significant roles in atmospheric processes, leading to the formation of air pollutants such as $PM_{2.5}$ and tropospheric ozone ($O_3$), which subsequently harm human health.**"

> L95 P4:

[Figure]

**Figure 1. Effects of agriculture and food systems on atmospheric chemistry and downstream impacts on human and ecosystem health, with direct linkages to various Sustainable Development Goals.**

*2) The manuscript, as it stands, focuses on the magnitude of contributions, but the basic processes influencing emissions and the chemistry pathways need to be explicitly mentioned. This is also true for the discussion of impacts - while there is extensive discussion of the magnitude of impacts, the biochemical pathways that enable such impacts are minimally discussed. This is important for a non-expert to readily understand the complexity and non-linearities in atmospheric chemistry.*

We acknowledge the reviewer's suggestion about the necessity to include the basic processes of nitrogen emissions from agriculture and food systems and their impacts on atmospheric particulate matter and ozone chemistry, especially to enhance understanding for non-experts readers.

We have revised the manuscript to include explanation of the key biochemical and atmospheric processes involved in emissions and their impacts in Section 2. As our paper is an opinion article rather than a comprehensive review, we have focused only on the most important aspects closely related to the discussion in subsequent sections. We aim to clarify the complexities and nonlinearities associated with agricultural emissions and their impacts. Additionally, we have cited the references that provide more detailed information on soil nitrogen emissions and their transformation in atmosphere, allowing readers interest in further details to explore these processes more thoroughly.

In Section 2.1.1 Sources, processes, and characteristics (Section 2.1 Emissions of reactive nitrogen)

[revised manuscript text omitted]

2. *Comparisons to typical sectors (power generation/industry/transportation) should be provided. Agriculture and food systems are typically unregulated but often have comparable if not outsized impacts on air pollution.*

We agree with the reviewer's suggestion to compare the contribution of agriculture and food systems with other well-regulated sources of air pollution. To address this, we have now extended the introductory part of Section 2.3, "Effects on atmospheric chemistry and ecosystems" to emphasize the relatively significant yet largely neglected roles of agriculture
       and food systems in air quality management.

       L230 P9: "**The significant roles that agriculture and food systems play in shaping
       atmospheric chemistry are increasingly realized. It is estimated that they contribute to
22–53% of PM2.5 and 5–25% of O3 pollution, which are contributions comparable to
       those of other well-regulated sources driven by fossil fuel combustion such as the energy
       and transportation sectors (Crippa et al., 2022a). However, there is still a lack of
       thorough investigation, particularly in underdeveloped regions such as Africa and South
       Asia. In this section, we highlight the latest findings along with the uncertainties and
limitations associated with the impacts of agriculture and food systems on the
       atmospheric environment.**"**

   3.  *The manuscript has briefly mentioned the need for life-cycle lens (para 75) to characterize the
       challenges with food systems. The recent literature and many conflicting evidence (as in the case
       of food miles) should be summarized and better highlighted.*

       We appreciate the reviewer's suggestions to deepen the discussion of the life-cycle
assessment of agriculture and food systems. Initially, we briefly mentioned the importance of
       the emissions from non-farm stages (L117 P5) and called for the full comprehensive
       investigation but did not sufficiently highlight life-cycle studies.

       L117 P5 "When considering the entire food systems beyond agricultural production, Nr
       emissions can be even higher. Food-system energy use, encompassing activities such as
fertilizer production, transportation, and processing, along with land use change driven by
       agricultural expansion, also contributes to substantial $NH_3$ and $NO_x$ emissions
       (Balasubramanian et al., 2021)."

       We have now expanded our discussion to briefly include these upstream and downstream
       emission estimates and their uncertainties, in the Section 2.1.2 "Emission estimates and
associated uncertainties". These uncertainties are often the main reasons for inconsistencies
       between studies.

       Furthermore, informed by the reviewers, we acknowledge conflicts in the literature about
       food miles (Li et al., 2022), particularly regarding the magnitude of emissions along the food
       chain and the implications of these studies for food trade or localization strategies. As this is
an opinion paper, we have succinctly summarized the characteristics of these studies and
       emphasized the need for further refinement in emission estimates. This focus aims to
       illuminate essence of the ongoing debates, rather than merely describing them.

       L182 P7 "**Along the entire food supply chain, emission estimation beyond the on-farm
       stage generally employs a similar EF method. Uncertainties associated with these
estimates primarily stem from the activities themselves, as well as from the
       corresponding EFs, due to the paucity of activity data. This issue is particularly
       profound in emissions originating from food transportation, which involves aspects such
       as transportation distances, means (e.g., road, rail, or ship), and refrigeration
       technology. International trade further complicates such estimation. Additional
uncertainties arise from how the boundaries of the food systems are defined; e.g., some
       studies considered the transportation of fertilizers, machinery and pesticides, while
       others did not (Li et al., 2022). A series of comprehensive assessment and life-cycle
       frameworks have been proposed recently to estimate global emissions from the entire
       agriculture and food systems (Crippa et al., 2022a; Li et al., 2022; Li et al., 2023).
However, these frameworks still suffer uncertainties in collecting activity data and**

**assuming different food trade policies, underscoring the need for further refinement in their methodology for emission estimation.**"

4.  *Sections 2.3.1-2.3.4: There is a lot of useful information here, but it may help with readability if the contents were structured in a similar blocks of information. Section 2.3.4, for example, has a lot of discussion of how to model the impacts but the content does not follow the thought indicated in the title.*

Thank you for your suggestions on streamlining the content in the discussion part to improve the quality of this manuscript. In Sect. 2, we focused on the four key issues on how agriculture and food systems affect atmospheric chemistry and ecosystems. The title of each subsection corresponds to specific pathways that drive the effects; for example, the impacts of agriculture on $PM_{2.5}$ are primarily due to $NH_3$ emissions, thus the title of this section reflects that focus. Similarly, the titles of other subsections are crafted in the same manner to emphasize their specific key issues.

More importantly, as this is an opinion article rather than a comprehensive review, we structured our discussion by first summarizing the current key understanding of each issue and then highlighting the associated uncertainties and gaps, which are mostly related to methodologies. Therefore, the content is structured in a similar manner across different sections, but the specific details discussed may vary, as the gaps associated with different topics are not the same, which might give the impression that these subsections are not similarly structured.

To enhance readability, we have added a sentence at the end of the introductory paragraph in Sect. 2 and Sect. 2.3 to inform readers about what is covered in the following sections.

L81 P3 "2 How agriculture food systems shape atmospheric chemistry and air pollution

Agriculture and food systems profoundly impact the atmosphere, most dominantly through the substantial emissions of reactive nitrogen (Nr) compounds from cropland and livestock systems… **Below is not meant to be a comprehensive review but is intended to highlight the key understanding, as well as the lack thereof, of the effects of agriculture and food systems on atmospheric chemistry. Figure 1 summarizes the important stages and impacts of agriculture and food systems via shaping atmospheric chemistry.**"

L216 P8 "2.3 Effects on atmospheric chemistry and ecosystems

Once released into atmosphere, agricultural and food emissions are actively involved in atmospheric processes…However, there is still a lack of thorough investigation, particularly in undeveloped regions such as Africa and South Asia. **In this section, we highlight the latest findings along with the uncertainties and limitations associated with the impacts of agriculture and food systems on the atmospheric environment.**"

5.  *A lot of the challenges in food systems are dietary choices (either personal choices or as shaped by larger environmental, economic, social structures). The discussion of these choices and impacts must be better highlighted. There is some discussion in Section 3.3. but it must include choices driven by diets (plant v/s animal rich), nutrition (triple burden of malnutrition), access - and of course food loss and waste as nicely discussed by the authors.*

We totally agree with the reviewer that dietary choices, along with food loss and waste, are key challenges in managing food systems. Our group is actively engaged in this field and recently published a keynote study on how dietary shifts to less meat-intensive could improve air quality in China. Upon reviewing your comments and revisiting Section 3.3, we recognize that while we have highlighted these aspects, their presentation sequence – initially focusing on food loss and waste before discussing dietary choices – may not be the most optimal.

In response, we have restructured the section to prioritize the discussion of dietary choices, reflecting their significant impacts on food system management. We have also introduced a new opening to underscore the importance of these topics. Additionally, we have enriched our discussion on dietary choices with latest literature and integrated the consideration of nutritional factors into this discussion.

L426 P15 "The entire food systems include not only on-farm production but also upstream and downstream stages such as agricultural input (e.g., fertilizer, pesticide) production, food processing, distribution, storage, retail and consumption. **Emission estimation and mitigation strategies for these off-farm stages as well as along the whole food chain are further complicated by dietary changes and food loss and waste, which can affect emissions at any stage along the whole food chain.** The widespread dietary shifts from plant-based to meat-intensive diets are the key driver for the globally increasing food demand, and meat-intensive diets are not only linked to increased risks of cardiovascular diseases, cancers, and type-2 diabetes, but also pose severe environmental threats (Gbd, 2019; Liu et al., 2021a). For instance, during 1980–2010 in China, dietary change alone could raise $NH_3$ emissions by 63% and annual mean $PM_{2.5}$ by up to ~10 $\mu g\,m^{-3}$ (Liu et al., 2021a). The study further suggested that adopting more sustainable, healthier, less meat-intensive diets could decrease annual mean $PM_{2.5}$ by 2–6 $\mu g\,m^{-3}$ in China. Likewise, a worldwide shift to plant-based diets could cut agricultural emissions significantly, by 44–86%, especially in regions with extensive livestock production (Springmann et al., 2023). Such dietary changes are expected to lower $PM_{2.5}$ and $O_3$ pollution by 3–7% and 2–4%, respectively, reduce premature mortality by 3–6%, and enhance economic output by 0.5–1.1%. **However, a recent detailed study on alternative dietary shifts argued that specific changes should be made cautiously, as some types of shifts aimed to improve health and nutrition may increase emissions (Guo et al., 2022). Dietary shifts toward a more plant-based diet, which encourage more intake of fruits, vegetables, and dairy products, can sometimes increase Nr emissions if such shifts require higher fertilizer inputs in low-NUE croplands.**

**In addition,** food loss typically occurs in the pre-production and production stages due to inadequate management and technology, whereas food waste happens during retail and consumption. About one third of the total food production (~1.3 billion tonnes) is discarded as food loss and waste (FLW) (Shafiee-Jood and Cai, 2016). Efforts to reduce FLW have shown promising results in mitigating $NH_3$ emissions and $PM_{2.5}$ pollution, with estimates suggesting a potential reduction of up to 11.5 Tg in $NH_3$ emissions and a decrease of about 5 $\mu g\,m^{-3}$ in $PM_{2.5}$ levels worldwide (Guo et al., 2023). **In relation to nutrition demand, populations with excessive calorie intake are recommended to shift their diets toward healthier nutritional patterns, which can also reduce FLW and emissions (Lopez Barrera and Hertel, 2023).**"

6. *Overall, the authors should consider being succinct as there is overlap in messaging across the manuscript.*

Thank you for your feedback. We have carefully reviewed our manuscript and streamlined the content by removing sentences that expressed similar ideas within the same sections. However, we have intentionally retained certain messages that recur across different sections. For example, some concepts are introduced briefly in the introduction to establish a foundation, then explored in greater detail in subsequent sections to discuss current understanding and identify knowledge gaps. These topics are further emphasized in our final section to summarize the key gaps and then propose how science can address each issue. This structure is designed to ensure clarity and coherence, providing convenience for readers who may be interested in specific aspects of our discussion.

L51 P2 "The global food systems, including all the stages of pre-production, production, post-production, consumption and waste management, are estimated to account for 58% of global anthropogenic emissions of... Such emissions are estimated to be responsible for 22% of global mortality arising from poor air quality and 1.4% of global crop production losses in year 2018 (Crippa et al., 2022b)…
The nitrogen load released…"

L142 P5 "Since agricultural emissions are influenced by multiple factors, including meteorological conditions, soil properties, and farming practices, the most advanced EF … can reflect the nonlinear responses of emissions to  their major drivers."

**Response to Referee #2**

*The authors have made a good effort to collect information about sources and effects of nitrogen related to the agriculture and food system. Although there is wide variety of issues that has been dealt with, the overall causal link between are mostly lacking. Especially for new readers in the field, these causal links could be better explained for a better understanding of the issues.*

We greatly appreciate your acknowledgement of our efforts to review and summarize studies on how agriculture and food systems influence atmospheric chemistry. Our article highlights the current understanding, knowledge gaps, and mitigation strategies regarding agricultural emissions and their adverse impacts. Since this is an opinion article rather than a comprehensive review, we have chosen to emphasize the most important issues, which may have led to a lack of detailed background information to fully construct the casual linkages from sources and emissions to impacts. We agree that strengthening these casual linkages would enhance the article's readability, especially for non-experts and new readers in this field, and also benefit public awareness of the importance of considering agriculture and food systems in air quality management and achieving sustainable development goals.

To improve the casual linkages of this opinion article, we have made the following enhancements:

1) Added an introductory paragraph at the beginning of Section 2, 'How agriculture and food systems shape atmospheric chemistry and air pollution', which succinctly delineate the sources, emissions, and impacts of agriculture and food systems.

L82 P3: "**Agriculture and food systems profoundly impact the atmosphere, most dominantly through the substantial emissions of reactive nitrogen (Nr) compounds from cropland and livestock systems, but also through other atmospheric pollutants such as primary particulate matter (PM), carbon monoxide (CO), methane ($CH_4$), $SO_2$, and VOCs via agricultural burning,**

**energy use of the whole food systems, and deforestation to clear lands for agriculture. Among these compounds, $NH_3$, $NO_x$, and HONO are inherently chemically active and play significant roles in atmospheric processes, leading to the formation of air pollutants such as $PM_{2.5}$ and tropospheric ozone ($O_3$), which subsequently harm human health.**"

2) Provided additional information about the underlying biogeochemical processes that drive emissions and the atmospheric processes by which the emissions transform into air pollution or are ultimately deposit back onto ecosystems.

In Section 2.1.1 Sources, processes, and characteristics (Section 2.1 Emissions of reactive nitrogen)

[revised manuscript text omitted]

*The authors mention knowledge gaps in their introduction. I suppose that most of these knowledge gaps are listed in the closing section. However, if these are indeed the knowledge gaps, they are*
*rather general without sometimes a clear link to the earlier sections. Furthermore, it needs to be clear that these knowledge gaps are most likely not general (global), but rather regional. This because most of the knowledge gaps are already addressed in different regions of the world. In that case these are rather information gaps that knowledge gaps (for which additional research is needed). I would suggest to make clear what is already done and where, so others can take notice and learn from it.*

Thank you for your insightful comments. As an opinion type article, our primary focus has been on identifying knowledge gaps related to agriculture and food systems research. Given the complexity of these systems, we separately discussed the different issues in each subsection of Sect. 2, by first outlining the current understanding and then detailing the knowledge gaps, thus integrating these gaps throughout Sect. 2 rather than consolidating them
in the closing section. The closing sections were designated to offer specific suggestions for each topic covered in Sect. 2, proposing further studies and discussing linkages to SDGs.

Upon reviewing the final section, we agree with the reviewer's comments that enhancing the visibility of these knowledge gaps will help readers, especially non-experts, understand this topic more clearly. In response, we have added summary sentences before the suggestions in
the closing section, corresponding to the gaps we highlighted in earlier subsections, to help readers grasp the key ideas of the gaps more clearly. We have now also included specific references to the parts where these gaps are discussed in detail (indicated in brackets).

Regarding the suggestions to regionalize the gaps, we partially agree with the reviewer's comments that "most of the knowledge gaps are already addressed in different regions of the
world". Even in developed countries, agriculture and food systems are not as thoroughly investigated as other sources, leading to significant uncertainties in emission estimation and their downstream impact assessment. In less developed regions such South Asia and Africa, these gaps are more evident due to considerably less attention and limited research resources compared to more developed regions. We have now emphasized this more and expanded our
discussion to highlight where addressing these gaps is the most urgent.

L87 P3: "… **Previous studies have enhanced our understanding of the mechanisms and driving factors behind agricultural emissions, allowing for improved evaluation of their impacts on air quality, human health and ecosystems (Butterbach-Bahl et al., 2013; Crippa et al., 2022a; Gu et al., 2023; Pilegaard, 2013). However, substantial uncertainties remain in these studies.** …"

L230 P9: "… **The significant roles that agriculture and food systems play in shaping atmospheric chemistry are increasingly realized. It is estimated that they contribute to 22–53% of PM$_{2.5}$ and 5–25% of O$_3$ pollution, which are contributions comparable to those of other well-regulated sources driven by fossil fuel combustion such as the energy and transportation sectors (Crippa et al., 2022a). However, there is still a lack of thorough investigation, particularly in underdeveloped regions such as Africa and South Asia.** …"

L469 P16: "The previous sections have highlighted how agriculture and food systems… To that end, as reviewed above **(Sect. 2.1), a better understanding of the magnitudes and drivers of Nr emissions is much needed,** and scientists need specifically to 1)…2)…3)…**These improvements would greatly help decrease uncertainties associated with Nr emission estimates and their adverse impacts on the atmospheric environment, and thus help us** devise better control policies… **Further improvements are still needed in Europe, China, and also the US where no specific mitigation targets have been planned, but relatively extensive research in these regions has already informed policy approaches elsewhere.** Other countries and regions are expected to follow suit, and more research for especially poorly researched, developing regions such as those in South Asia, Southeast Asia, Africa and Latin America are necessary to guide their mitigation efforts."

L498 P17: "To mitigate **agricultural and** food-system emissions of Nr and other pollutants, **in light of the complex region- and species-specific responses of Nr emissions across multiple stages from the whole food systems (Sect. 3)**, we also need to focus more research efforts on the various mitigation pathways, including: 1) …2) …3) …4) … **Such efforts are recommended for both developing and developed regions.**"

L517 P18: "… atmospheric scientists are necessary to better quantify the ecosystem input of Nr via atmospheric deposition **(Sect. 2.3.4)**, especially via (Zhang et al., 2021b):1) …2) … 3) …4) …"

*What furthermore is missing (I think), is the notice that the linkage between the agriculture/food systems and the SDG's is not only through the atmospheric pathway, but also (and sometimes maybe even more) through the aquatic pathway. I would suggest to, at least, mention this and, when possible, to give an indication of the relative contributions from these two pathways. This to provide some more context to the reader.*

Thank you for your insightful comment. We fully agree that the linkages between agriculture and food systems and the SDGs involve not only atmospheric pathways, which predominantly affect atmospheric chemistry through nitrogen emissions, but also aquatic pathways. These aquatic pathways impact the balance of aquatic systems by releasing nutrient compounds, especially nitrate, into open waters such as rivers, lakes, and ocean, leading to decreased water quality, eutrophication, and damage to biodiversity. These aquatic pathways are important to SDG 3 Good Health and Well-being, SDG 6 Clean Water and Sanitation, and SDG 14 Life below Water, as they are directly related to water quality and the balance of aquatic ecosystems.

As reviewer suggested, although our opinion article primarily focuses on the atmospheric chemistry in achieving the SDGs, we have added in various places of our manuscript to highlight the importance of aquatic pathways and the impacts of agriculture on aquatic ecosystems, including the linkage to SDG 6 and SDG 14:

L56 P2 "Moreover, reactive nitrogen (Nr) compounds of agricultural origins including $NH_3$, $NO_x$, nitrous acid (HONO) and their reaction products, can readily be deposited back onto the land surface **and waterbodies,** causing various effects on terrestrial **and aquatic** ecosystems, including more serious nutrient leaching, soil acidification (Guo et al., 2010; Lu et al., 2011), **and eutrophication (Deng et al., 2023; Deng et al., 2024a; Jickells et al., 2017; Liu et al.,**
**2023b)**."

L339 P12 "2.3.4 Impacts of nitrogen deposition on terrestrial and **aquatic** ecosystems

The Nr compounds of agricultural origins often undergo transport and chemical transformation, and are eventually deposited back onto the surface of terrestrial and aquatic ecosystems, resulting in increased nitrification, nutrient leaching, soil acidification (Guo et
al., 2010), **eutrophication (Liu et al., 2023b)**, and biodiversity loss (Simkin et al., 2016), while also possibly enhancing forest growth **and** carbon storage (Liu et al., 2022; Lu et al., 2021a; Quinn Thomas et al., 2010) **as well as marine productivity (Jickells and Moore, 2015). Enhanced Nr deposition to the open ocean has been known to generate high-productivity, low-oxygen zones with disrupted ecosystem functions (Doney, 2010)** Due
to historically more stringent emission controls on combustion $NO_x$ than agricultural $NH_3$ emissions, Nr deposition patterns are shifting from being nitrate-dominated to ammonium-dominated, a trend observed in the US and China, and expected in Europe (Chen et al., 2020; Li et al., 2016; Liu et al., 2020; Tan et al., 2020)**, not only over inland but also in coastal areas (Liu et al., 2023b)**."

L510 P18 "… Furthermore, **agriculture influences ecosystems not only via atmospheric Nr deposition but also via direct nutrient leaching and runoff to waterbodies. Nitrogen pollution can bring tremendous disruptions to terrestrial and aquatic ecosystems, often modifying both ecosystem productivity and biodiversity. Therefore, mitigating agricultural and food-system emissions also helps** us strive toward **SDG 14 "Life Below**
**Water", which aims to conserve marine and coastal ecosystems, and sustainably use their resources for sustainable development, and** SDG 15 "Life on Land", …"

Connection to water quality SDG 6 has already been emphasized in the context of socioeconomic impacts:

L528 P18: "… Often reducing food-system emissions would bring immediate health benefits
to the farmers and people in agricultural regions in general due to the reduced exposure to airborne and waterborne (e.g., fertilizer, pesticide and animal waste runoffs) agricultural pollutants, which would in the long term improve their productivity and livelihood. … SDG 6 "Clean Water and Sanitation", which aims to ensure universal access to clean water and sanitation, improve water quality and promote sustainable water management practices; …"

We have opted to not focus on SDG 6 as one of the main goals (as in Fig. 1) because atmospheric deposition per se is generally not a major source of water pollutants that are relevant for drinking water quality, but certainly reducing agricultural Nr emissions will help improve water quality as a co-benefit.

**Response to Community Referee**

*This opinion reviews agricultural and food system emissions of Nr and other atmospherically relevant compounds, their fates and impacts on air quality, human health, and terrestrial ecosystems, and how such emissions can be potentially mitigated through better cropland management, livestock*
*management, and whole food-system transformation. In general, this paper is well-organized and written. I have minor comments to strengthen the paper before it can be published in ACP.*

Thank you for your positive feedback and constructive suggestions on our opinion article. We greatly appreciate your insights, and the corresponding references you provided were instrumental in guiding our revisions. We have carefully considered each of your suggestions
and made appropriate revisions accordingly.

*First, I am missing the impact of N deposition on the oceanic N cycle and ecosystems, which I think is a very important part. Food-driven N deposition poses a significant part of human N input to oceans. See refs: Liu et al., Modeling global oceanic nitrogen deposition from food systems and its mitigation potential by reducing overuse of fertilizers, https://doi.org/10.1073/pnas.2221459120; Jickells et al.,*
*A re-evaluation of the magnitude and impacts of anthropogenic nitrogen inputs on the ocean. Global Biogeochem. Cycl. 31, 289–305 (2017).*

We have expanded our discussion to include N deposition on open waters and its impact on aquatic systems, with citations of the references you suggested.

L56 P2 "Moreover, reactive nitrogen (Nr) compounds of agricultural origins including $NH_3$,
$NO_x$, nitrous acid (HONO) and their reaction products, can readily be deposited back onto the land surface **and waterbodies,** causing various effects on terrestrial **and aquatic** ecosystems, including more serious nutrient leaching, soil acidification (Guo et al., 2010; Lu et al., 2011), **and eutrophication (Deng et al., 2023; Deng et al., 2024a; Jickells et al., 2017; Liu et al., 2023b).**"

L339 P12 "2.3.4 Impacts of nitrogen deposition on terrestrial and **aquatic** ecosystems

The Nr compounds of agricultural origins often undergo transport and chemical transformation, and are eventually deposited back onto the surface of terrestrial and aquatic ecosystems, resulting in increased nitrification, nutrient leaching, soil acidification (Guo et al., 2010), **eutrophication (Liu et al., 2023b)**, and biodiversity loss (Simkin et al., 2016),
while also possibly enhancing forest growth **and** carbon storage (Liu et al., 2022; Lu et al., 2021a; Quinn Thomas et al., 2010) **as well as marine productivity (Jickells and Moore, 2015). Enhanced Nr deposition to the open ocean has been known to generate high-productivity, low-oxygen zones with disrupted ecosystem functions (Doney, 2010) .** Due to historically more stringent emission controls on combustion $NO_x$ than agricultural $NH_3$
emissions, Nr deposition patterns are shifting from being nitrate-dominated to ammonium-dominated, a trend observed in the US and China, and expected in Europe (Chen et al., 2020; Li et al., 2016; Liu et al., 2020; Tan et al., 2020)**, not only over inland but also in coastal areas (Liu et al., 2023b)**."

L510 P18 "… Furthermore, **agriculture influences ecosystems not only via atmospheric**
**Nr deposition but also via direct nutrient leaching and runoff to waterbodies. Nitrogen pollution can bring tremendous disruptions to terrestrial and aquatic ecosystems, often modifying both ecosystem productivity and biodiversity. Therefore, mitigating agricultural and food-system emissions also helps** us strive toward **SDG 14 "Life Below Water", which aims to conserve marine and coastal ecosystems, and sustainably use**
**their resources for sustainable development, and** SDG 15 "Life on Land", …"

*Second, I suggest a) the authors split the agricultural NH₃ emissions into crop and livestock emissions, and b) confirm all numbers in Table 2 (whether it is total agricultural NH₃ emissions or just part of them) since I have seen a huge gap between different studies (26-60 Tg N yr⁻¹). c) There are also recent studies reporting global NH₃ emissions which should be cited properly. See refs: Liu et al., Exploring global changes in agricultural ammonia emissions and their contribution to nitrogen deposition since 1980 https://doi.org/10.1073/pnas.2121998119; Yang et al., Improved global agricultural crop- and animal-specific ammonia emissions during 1961–2018, https://doi.org/10.1016/j.agee.2022.108289*

(a) In Table 2, we summarized the latest global estimations of $NH_3$ and $NO_x$ emissions from agriculture and total emissions from all sources to emphasize the significant roles of agriculture. As most of these studies did not specify the magnitudes of emissions from croplands and livestock separately, we did not include such a breakdown in the table. However, in our main text, we discussed the differences in estimating $NH_3$ emissions from croplands and livestock. In response to your feedback, we have added a sentence describing the contributions of $NH_3$ from cropland and livestock based on a global study.

L106 P4 "Consequently, in both cropland and livestock systems, a significant portion of the added nitrogen is lost... **Globally, around agricultural 60% of $NH_3$ emissions are from livestock production, with the rest from cropland systems (Yang et al., 2023a)**."

(b) Thank you for your careful examination of the data. However, it appears there has been a slight mix-up; the agricultural $NH_3$ emissions you mentioned are in Table 1, not Table 2.

We have verified the numbers in Table 1 and confirm their accuracy. The discrepancies between different studies primarily stem from the uncertainties inherent in their methodologies, particularly regarding the emission factors employed. Our article includes detailed discussion of the various estimates among studies and their underlying reasons. Additionally, discrepancies can also be attributed to the use of activity data from different baseline years (as indicated in Table 1, column "Base year"). These years often correspond to various environmental and socioeconomic conditions, which can significantly influence agricultural emissions and, consequently, the resulting estimates. We have now included a discussion of the driving factors of agriculture emissions in revised article as follows.

Upon reviewing the studies listed in Table 1, we noted that one study we cited had a notably older baseline year, 2010, which is approximately 10 years earlier than the others (2010–2018). We have thus removed this study from the Table 1, as the differences for this study compared to others may be caused by the driving factors rather than the methodologies discussed in the main text.

L121 P5 "**Agricultural and food-system Nr emissions exhibit high spatiotemporal variations, responding nonlinearly to meteorological conditions, soil properties, and farming practices, influenced by microbial activities. Typically, regions with intensive fertilizer use and low nitrogen use efficiency (NUE, i.e., the ratio of nitrogen removed with the harvest to nitrogen input) tend to have the highest emission levels. High temperature and precipitation also contribute to increased emissions and modulate their interannual variability (Griffis et al., 2017; Shen et al., 2020). NUE and Nr emission changes can be further driven by socioeconomic factors, with divergent patterns in different countries depending on the population level, economic growth, farm size, urbanization level, international trade, and their interactions. A series of global-scale, long-term analyses have suggested that developed regions with well-managed urban-rural development tend to have lower agricultural Nr emissions as their large-scale farming along with advanced**

**agricultural technology and coupled cropland-livestock systems can enhance NUE and maintain agricultural productivity to support international trade (Deng et al., 2024b; Gu et al., 2020; Ren et al., 2022; Liu, 2023). In the future, fertilizer input is expected to further increase to feed the growing global population, potentially further elevating Nr emissions if not efficiently managed. Meanwhile, climate change has been estimated to increase Nr emissions by ~80% between 2011 and 2100, and the resulting more frequent extreme weather events may induce extensive dry-wet and freeze-thaw cycles that can further exacerbate such increases (Griffis et al., 2017; Shen et al., 2020; Wagner-Riddle et al., 2017)."**

L180 P7:

Table 1. Global estimates of $NH_3$ and $NO_x$ emissions (Tg N yr$^{-1}$).

| Sources | Method | Base year | Agricultural NH$_3$ | Total NH$_3$ | Agricultural NO$_x$ | Total NO$_x$ |
|---|---|---|---|---|---|---|
| EDGAR (Crippa et al., 2018) | Bottom-up | 2018 | 38.2 | 43.7 | 1.9 | 36.5 |
| CEDS (Mcduffie et al., 2020) | Bottom-up | 2017 | 39.2 | 51.6 | 2.3 | 37.7 |
| HTAP (Crippa et al., 2023) | Bottom-up | 2018 | 42.5 | 48.5 | 1.7 | 35.6 |
| Fowler et al. (2013) | Bottom-up | 2010 | 59.9 | 69 | | |
| Yang et al. (2023b) | Bottom-up | 2018 | 60 | | | |
|  |  |  |  | | | |
| Huang et al. (2017) | Bottom-up | 2014 | | | | 39.2 |
| Luo et al. (2022b) (EDGAR as prior) | Top-down | 2018 | | 71.9 | | |
| Miyazaki et al. (2017) (EDGAR as prior) | Top-down | 2014 | | | | 47.5 |

(c) Upon reviewing of our manuscript, we confirmed that we have cited the two publications you recommended.

*Third, are there any social-economic drivers for changes in food emissions? For instance, when we talked about NH$_3$, it's usually controlled by increasing population and food production (N fertilizer, livestock). NH$_3$ changes are mainly affected by temperature and fertilizer applications. I hope to see some additional discussions on long-term changes and their social-economic drivers. How the urbanization affect emissions and pollution? (see refs: L. Liu. 2023 Nature, https://doi.org/10.1038/d41586-023-02753-9; Deng et al. 2024 Nature communications, https://doi.org/10.1038/s41467-023-44685-y).*

We agree that the agriculture and food emissions are influenced by socioeconomic factors, in addition to environmental factors. In response, we have expanded our discussion on how socioeconomic factors drive emission changes, with citations of the references you recommended. See our response to point (b) above.

*Fourth, I would like to see some discussion on how climate change/extreme weather affects food*
*emissions and production, which I think is an essential part of future efforts on maintaining food*
*production. Please see refs: Liu et al, China's response to extreme weather events must be long term,*
*https://doi.org/10.1038/s43016-023-00892-w; Lesk, C. et al. Nat. Rev. Earth Environ. 3, 872–889*
*(2022).*

We have expanded our discussion to include the impacts of climate change and extreme
weather events on agricultural emissions. However, we only briefly mention how these
changes exert pressure on crop yields, as our primary focus is on agricultural emissions. Also, we have cited the recommended references.

L72 P3 "… But how can we do that without compromising the needs of people to be food-secured and nourished? **How can we achieve these multiple goals under the concurrent threat of climate change, which can both impair crop production and elevate**
**agricultural emissions?** Here we argue that, … how these compounds are transported, transformed and deposited back onto the surface, **how all these processes are sensitive to climate change,** and how the food systems can be modified in technologically feasible, economically viable and socially equitable manners to abate emissions."

See also the additional text cited in our response to point (b) above.

---

## Author Response (AR2)

Author Responses to Referee's Comments on **"Opinion: Understanding the impacts of agriculture and food systems on atmospheric chemistry is instrumental to achieving multiple Sustainable Development Goals"** by Amos P. K. Tai et al. (MS No.: egusphere-2024-293)

We would like to thank the reviewer for the positive feedback and for acknowledging our efforts in addressing the concerns raised during the previous round of revisions. We have carefully addressed the remaining points, as outlined below, and our point-by-point responses are provided below. The referee's comments are *italicized*, our new/modified text is highlighted in **bold**.

**Response to Referee**

*1. At several places, you write 'atmospheric chemistry' but it is not clear whether you mean 'atmospheric composition' or 'chemical processes in the atmosphere' - or both. Please be more specific where possible.*

Thank you for your suggestion on clarifying the use of "atmospheric chemistry" in our article. We have revised the text to specify whether we are referring to "atmospheric composition", "chemical processes in the atmosphere", or both, depending on the context. For the use of "atmospheric chemistry" in the title and some summary sentences, we have retained the term, as it conveys the broader meaning intended. When both senses are intended or when the use of the term is intended to be more general, we also retain the use of "atmospheric chemistry".

L16 P1 "With that, we highlight important knowledge gaps that warrant more extensive research, and argue that we scientists need to provide a more detailed, process-based understanding of the impacts of agriculture and food systems on atmospheric chemistry, **including both chemical composition and processes**, especially as the importance of emissions from other fossil fuel-intensive sectors is fading in the face of regulatory measures worldwide."

L32 P2 "Such momentum gathered is arguably also a promising development for air quality managers and policy makers worldwide, because agriculture and food systems are major sources of various short-lived chemical species that shape  **chemical composition and processes in the atmosphere, which in turn** contribute to air pollution."

L68 P3 "All these findings highlight the importance of agriculture and food systems in shaping atmospheric  **composition and chemical processes**, **as well as** air pollution and the associated public health and ecosystem impacts."

L97 P3 "Below is not meant to be a comprehensive review but is intended to highlight the key understanding, as well as the lack thereof, of the effects of agriculture and food systems on atmospheric **composition and chemical processes**."

L177 P6 "Another limitation is … in agricultural $NH_3$ emissions on atmospheric  **composition**, as $NH_3$ typically peaks within several days after fertilizer application (Nelson et al., 2019)."

L242 P9 "The significant roles that agriculture and food systems play in shaping **chemical processes in the atmosphere** are increasingly realized."

L344 P12 "Finally, beyond $NH_3$ and $NO_x$, agricultural emissions of HONO are also important for atmospheric chemistry **by affecting chemical processes in the atmosphere**, mostly because of its photolysis product, hydroxyl radical (OH), the primary oxidant in the troposphere, which is heavily involved in $PM_{2.5}$ and $O_3$ chemistry (Oswald et al., 2013)."

L480 P17 "However, the SDGs are not meant to be standalone objectives, … agricultural and food-system contributions to atmospheric chemistry, **including both the composition and chemical processes of the atmosphere,** is indeed crucial to help stakeholders achieve SDG 2 in synchrony with other SDGs, especially SDG 3 "Good Healthy and Well-being", SDG 13 "Climate Action", SDG 14 "Life Below Water", and SDG 15 "Life on Land", but also various others more indirectly."

*2. l. 29: 'managers' misspelled.*

Thank you for bringing this typo to our attention. We have corrected the spelling of "managers" accordingly.

L 29–31 P2 "Such momentum gathered is arguably also a promising development for air quality  **managers** and policy makers worldwide, because agriculture and food systems are major sources of various short-lived chemical species that shape atmospheric chemistry and contribute to air pollution."

*3. l. 543: You may want to add 'aquatic' here as well.*

Thank you for this valuable suggestion to further enhance the sentence by including "aquatic" to comprehensively address the impacts of agricultural and food systems on both terrestrial and aquatic ecosystems. We have revised the sentence accordingly to better reflect the broader environmental impacts.

L 556 P19 We therefore opine that, in consideration of the substantial impacts of agricultural and food-system emissions on atmospheric chemistry, air pollution and subsequently on terrestrial **and aquatic** ecosystems, we as a society … become economically, socially and environmentally sustainable.